# Three New Compounds, Licopyranol A–C, Together with Eighteen Known Compounds Isolated from *Glycyrrhiza glabra* L. and Their Antitumor Activities

**DOI:** 10.3390/metabo12100896

**Published:** 2022-09-23

**Authors:** Shanshan Wang, Jameel Hizam Alafifi, Qin Chen, Xue Shen, Chunmei Bi, Yangyang Wu, Yihan Jiang, Yanan Liu, Yimeng Li, Dian He, Zhigang Yang

**Affiliations:** 1School of Pharmacy, Lanzhou University, Lanzhou 730000, China; 2Collaborative Innovation Center for Northwestern Chinese Medicine, Lanzhou University, Lanzhou 730000, China; 3State Key Laboratory of Applied Organic Chemistry, Lanzhou University, Lanzhou 730000, China

**Keywords:** *Glycyrrhiza glabra* L., licorice, flavonoid, coumarin, glycyrol, antitumor, RKO cells

## Abstract

*Glycyrrhiza glabra* L., known as licorice, is one of the most famous herbs in the world. In this study, we investigated the phytochemical and antitumor activities of *G. glabra*, especially its anti-colorectal cancer activities. *G. glabra* was extracted with 70% methanol, and the ethyl acetate layer was separated by silica gel, ODS, LH-20 column chromatography, and semi-preparative HPLC to obtain the compounds. The structures were determined by NMR and MS methods. Three new compounds named licopyranol A–C (**1**–**3**), and eighteen known compounds (**4**–**21**) were isolated. Compounds with an isoprenyl group or dimethylpyran ring showed better antitumor activities. Licopyranol A (**1**) and glycyrol (**5**) both inhibited the proliferation, reduced clone formation and promoted apoptosis of RKO cells. The Western blotting assays showed that glycyrol significantly reduced the expression of E-cadherin, β-catenin, c-Myc, and GSK-3β proteins in RKO cells, suggesting that glycyrol may inhibit the growth of colorectal cancer RKO cells via the Wnt/β-catenin signaling pathway.

## 1. Introduction

The genus of *Glycyrrhiza*, which belongs to the Fabaceae family, consists of about 20 species from all over the world [1]. Among these species, *Glycyrrhiza uralensis* Fisch, *Glycyrrhiza inflata* Bat. and *Glycyrrhiza glabra* L. are used as licorice (*Gancao*) in traditional Chinese medicine to treat various diseases, such as cough, influenza, diabetes, and cancer, etc. *G. glabra* is also one of the most commonly used herbs in the world, which possesses hepatoprotective, anti-inflammatory, neuroprotective, antioxidant, and antiviral activities [2]. Phytochemical studies have revealed that *G. glabra* contains numerous phytochemicals, such as saponins, flavonoids, polysaccharides, and coumarins [3]. The prenylated flavonoids have received extensive attention because of their better biological activities [4]. Licorice is a natural source of prenylated compounds [5]. It has been reported that prenylated flavonoids from *G. uralensis* induced differentiation of B16-F10 melanoma cells [6]. In addition, glycybridin D (10 mg/kg, ip) decreased tumor mass by 39.7% in a mouse model established by an A549 human lung carcinoma xenograft [2].

The medicinal plants and natural ingredients for cancer treatments have attracted increasing interest [7]. Licorice is currently one of the main ingredients used for inflammatory diseases and cancer in traditional Chinese medicines. Most of the drugs that are used in cancer therapy have some serious side effects. Therefore, *Glycyrrhiza glabra*, and other natural anticancer agents have been extensively studied due to the fact that these compounds are considered to have better bioactivities and may improve the side effects of cancer therapy [8]. In recent years, biological activities, especially antitumor and cytotoxic properties of licorice extracts and their isolated compounds, have received much attention. The bioactive components of licorice have shown antitumor properties in both in vivo and in vitro studies [9]. It has been reported that licorice extracts exhibited cytotoxic effects on various cancer cells and 70% methanol licorice extract inhibited the proliferation of human breast cancer cell MCF7 and hepatocellular carcinoma cell HepG2 [10,11]. Flavonoids of licorice showed important inhibitory effects on colorectal, breast, prostate, liver, stomach, bladder, and lung cancers [12]. Furthermore, the isolated compounds from the methanolic and ethyl acetate extract of *G. glabra* were found to show significant cytotoxic and anticancer properties [13,14].

In this study, the extracts of *G. glabra* showed inhibitory activities on RKO and HT-29 cells and the ethyl acetate extract had better effects than the other layers at 20 μg/mL (Appendix A). As a result, 3 new compounds and 18 known compounds were isolated from *G. glabra*, and their inhibitory effects on 6 tumor cells were tested. Several compounds showed better antitumor activities. In addition, we further investigated the mechanism of inhibitory effects of licopyranol C (**1**) and glycyrol (**5**) on RKO cells. The results from this study are expected to provide lead compounds for cancer therapy in the future.

## 2. Materials and Methods

### 2.1. General Experimental Procedures

Ultrasonic extractor (Ningbo Scientz Biotechnology Co., Ltd) UV spectra were recorded on a Shimadzu UV-2500 spectrophotometer. NMR spectra at 400 MHz for ^1^H and 100 MHz for ^13^C were obtained with a Bruker AVANCE III-400 spectrometer in DMSO-*d*_6_, with TMS as the internal reference. HR-ESI-MS spectra were determined by an ultra-performance liquid chromatography ion mobility time-of-flight mass spectrometer (Agilent 1290 × 1290 − 6560 Ion Mobility QTOF-MS). Semi-preparative HPLC was performed on an EasySep-1050 liquid chromatograph, equipped with a Cosmosil 5C_18_-MS II column (10 × 250 mm, 10 μm). Silica gel (200–300 mesh; Qingdao Haiyang Chemical Co., Ltd., Qingdao, China), C_18_ reversed-phase silica gel (75C_18_-OPN, 50 μm, Nacalai Tesque, Inc., Kyoto, Japan), and Sephadex LH-20 (Pharmacia Biotek, Copenhagen, Denmark) were used for the column chromatography. Thin layer chromatography (TLC) was used with silica gel GF254 (Yantai Chemical Industry Research Institute, Yantai, China), and RP-18 F_254_s (Merck KGaA, Darmstadt, Germany).

### 2.2. Plant Material

The roots and rhizomes of *G. glabra* L. were collected in September 2020 in Xinjiang Autonomous Region, China, which was identified by Zhigang Yang. The voucher specimen (No. 202009001) was stored in the School of Pharmacy, Lanzhou University.

### 2.3. Extraction and Isolation

The ultrasound extraction method has become more popular due to its various features, such as low energy consumption, less extraction time, less active compound degradation, its suitability for thermo-sensitive compounds and high extraction yield [15]. This method has been applied previously for the isolation of secondary metabolites from licorice [16]. Thus, the dried roots and rhizomes (10 kg) were broken to pieces and extracted with 70% methanol 4 times, by soaking overnight and ultrasonic extraction 1 h under 45 kHz at 45 °C; then, the filtrate was combined, concentrated to dryness, and we obtained 966 g of the extract. The extract was diffused in water and successively extracted by petroleum ether, dichloromethane, ethyl acetate and *n*-butanol, respectively. The ethyl acetate extract (274 g) was applied over a silica gel column, then washed by petroleum ether-ethyl acetate (100:0-0:100, *v*/*v*) to acquire fractions A-I.

Fraction C (20 g) was subjected to silica gel column chromatography and washed by petroleum and ether-ethyl acetate to obtain fractions C1-C6. Fraction C3 was loaded over a silica gel column, then washed with CH_2_Cl_2_-MeOH (100:0 to 0:100, *v*/*v*) to obtain fractions C3A-C3F. Fraction C3B was loaded onto ODS column chromatography and then purified by PTLC to yield **1** (4.7 mg) and **2** (7.8 mg). Fraction C3C was chromatographed in the ODS column and washed with MeOH-H_2_O (1:1 to 1:0, *v*/*v*), and then purified by PTLC to yield **3** (5.7 mg), **4** (13.5 mg), **6** (12.2 mg), **12** (5.7 mg), **13** (57.8 mg), **16** (38.9 mg), **17** (7.8 mg) and **20** (11.3 mg).

Fraction D (24 g) was loaded onto the silica gel column and washed with petroleum and ether-ethyl acetate to obtain fractions D1-D6. Fraction D2 was further separated using the silica gel column, and washed with CH_2_Cl_2_-MeOH (100:0 to 50:50, *v*/*v*) to give fractions D2A-D2H. Fraction D2B was chromatographed on semi-preparative HPLC through MeOH-H_2_O (72:28, *v*/*v*, 4 mL/min) to provide compound **18** (5.7 mg), **19** (12.5 mg) and **21** (3.6 mg). Fraction D2C was isolated through Sephadex LH-20 column chromatography using CH_2_Cl_2_-MeOH (50:50, *v*/*v*) and was then re-chromatographed over semi-preparative HPLC using MeOH-H_2_O (70:30, *v*/*v*, 4 mL/min) to acquire compound **10** (13.4 mg). Fraction D4 was fractionated by silica gel column chromatography, using CH_2_Cl_2_-MeOH (1:0 to 1:1, *v*/*v*), and afforded fractions D4A-D4G. Fraction D4B was purified using Sephadex LH-20 column chromatography and washed using CH_2_Cl_2_-MeOH (1:1, *v*/*v*), then chromatographed by semi-preparative HPLC, using MeOH-H_2_O (70:30, *v*/*v*, 4 mL/min) to produce compound **14** (13.3 mg). Fraction D4C was purified by the Sephadex LH-20 column, washed with CH_2_Cl_2_-MeOH (50:50 to 0:100, *v*/*v*), then re-chromatographed on semi-preparative HPLC, using MeOH-H_2_O (75:25, *v*/*v*, 4 mL/min) to provide compound **8** (12.4 mg), **9** (13.1 mg) and **15** (21.4 mg)**.** Fraction D4D was chromatographed with Sephadex LH-20 column chromatography using CH_2_Cl_2_-MeOH (50:50 to 0:100, *v*/*v*), then re-chromatographed on semi-preparative HPLC, using MeOH-H_2_O (75:25, *v*/*v*, 4 mL/min) to acquire compound **5** (4.7 mg)**, 7** (5.5 mg) and **11** (13.5 mg).

### 2.4. Cell lines and Culture

Human colorectal cancer cells RKO were purchased from Wuhan Procell Life Sciences Co. Other cells were obtained from the National Infrastructure of Cell Line Resource. All cells were cultured with medium-augmented 10% fetal bovine clear fluid (serum) and 1% double antibodies (penicillin and streptomycin) under a standard incubation temperature at 37 °C in 5% CO_2_. The type of basal medium used for RKO cells was MEM medium; HT-29, HCT-116 cells and gastric cancer MGC-803 cells were cultured in RPMI-1640 medium; lung cancer A549 cells and liver cancer Huh7 cells were cultured in DMEM medium.

### 2.5. Cell Viability Assay

The cells were digested with trypsin and resuspended, inoculated into 96-well dishes with a density of (5 × 10^3^ per well) and incubated overnight. After the cells were plastered and incubated with complete medium containing various drug concentrations for 48 h, 20 μL of 5 mg/mL MTT solution was injected into each well and cultured in a constant temperature incubator for 4 h at 37 ^o^C; then, the supernatant was disposed of and 150 μL of DMSO was added, agitated for 10 min and at 570 nm, the absorbance was measured by an enzyme marker. Based on the formula below, the percentage of cell viability was calculated: viability percentage of cells (%) = ((OD_drug_ − OD_blank_)/(OD_control_ − OD_blank_)) × 100%.

### 2.6. Cell Proliferation Assay

To further investigate the inhibitory effects of the compounds, we designed a multiple concentration-time dependent assay. The RKO cells were incubated with complete medium containing different drug concentrations (10, 20, 40 μM) for 0, 24, 48 and 72 h, respectively. Then, 20 μL of 5 mg/mL MTT solution was injected into all the wells and after 4 h of incubation, the supernatant was removed and 150 μL of DMSO solution was added to all the wells, then the growth curve was plotted, using their absorbance at 570 nm.

### 2.7. Cell Cloning Assay

The RKO cells were digested with trypsin and resuspended; at a density of 1 × 10^3^ cells per well, they were inoculated in 6-well dishes and incubated overnight. After the cells were plastered, they were incubated with complete medium containing different drug concentrations for 7 days, and the complete medium was changed once every 3 days. At the end of the culture, the supernatant was disposed of, the cells were washed using PBS, affixed using 4% paraformaldehyde for half an hour, cleaned three times using PBS, then dyed by 0.1% crystalline violet for 10 min, cleaned with PBS to remove the unbound crystalline violet, then photographed and recorded.

### 2.8. Cell Cycle Assay

For cell cycle experiments, RKO cells at the logarithmic growth stage were digested with trypsin and resuspended, at a density of 1 × 10^6^ cells per well, then inoculated in 6-well dishes and cultivated overnight. After being plastered, they were incubated separately for 48 h in complete medium containing different concentrations of the drug. After washing with pre-chilled PBS and digestion with trypsin at the end of the culture, the cells were centrifuged at 200× *g* for 5 min, resuspended with pre-chilled 70% ethanol and fixed at 4 °C overnight. After washing with PBS, 100 μL RNase A solution was incorporated, then the cells were resuspended for 30 min at 37 °C in a water bath; 400 μL PI dying solution was incorporated and mixed, nurtured for 30 min at 4 °C and protected from light for 30 min at 4 °C. The results were detected and analyzed by a CytoFLEX flow cytometer and processed with Modfit LT 5.0.

### 2.9. Cell Apoptosis Assay

The RKO cells were digested with trypsin and resuspended in 6-well dishes at a density of 1 × 10^6^ per well and cultured overnight. Then, the cells were plastered and they were incubated in complete medium containing different drug concentrations for 48 h. After the incubation, the supernatant was discarded, the cells were digested with trypsin and collected, washed with PBS, resuspended by adding 1 mL of pre-prepared binding buffer, centrifugated at 300× *g* for 10 min, the supernatant was removed, 1 mL of binding buffer was added and the cells were resuspended to reach a density of 1 × 10^6^ cells/mL. After adding 100 μL of cells to each tube, 5 μL of Annexin V-FITC was incorporated into the tube, mixed gently and nurtured for 10 min without light, then 5 μL of PI solution was added and incubated for 5 min away from light; after completion, we added PBS to 500 μL, mixed gently, assayed and analyzed the results with a CytoFLEX flow cytometry analyzer.

### 2.10. Western Blotting Assay

The RKO cells were digested with trypsin and resuspended in 6-well dishes at a density of 1 × 10^6^ per well and cultivated overnight. After the cells were plastered, they were incubated with complete medium containing different drug concentrations for 48 h. After finishing the culture, the supernatant was removed, the cells were dissolved using trypsin and collected by centrifugation at 130× *g* for 3 min, the supernatant was disposed of and washed three times by PBS; the cells were lysed by adding pre-configured cell lysis solution, containing 1% PMSF and phosphatase inhibitor, in an ice bath for half an hour, centrifugated at 1300× *g* for 5 min and the supernatant was carefully aspirated for BCA protein quantification. After quantification, we added the load buffer and heated the solution in a metallic bath for 10 min at 100 °C.

Based on the results of BCA protein quantification, 25–30 μg of protein was added to each lane at once and the corresponding marker was added; the voltage was adjusted and SDS-polyacrylamide gel electrophoresis was performed. The gel containing the target and GAPDH protein bands was cut off, then, the gel and PVDF membrane were sandwiched between a cellulose pad and filter paper using a sandwich structure, which was deflated and then placed in a transfer tank and transferred into an ice bath with pre-prepared transfer solution. PVDF membranes were soaked by TBST solution that comprised 5% non-fat milk powder at 25 °C for 2 h. The sealed PVDF membranes were washed 3 times with TBST solution and incubated overnight at 4 °C with a solution containing primary antibodies (diluted 1:1000 with TBST) and the corresponding protein bands in an antibody incubator; after 3 washes with TBST solution, secondary antibodies (diluted 1:10000 with TBST) were incubated for 2 h and washed 3 times with TBST. The above films were developed with the ELC color development system, the X-ray film was pressed, the film was scanned and the film grey values were analyzed with image-pro plus.

### 2.11. Statistical Analysis

GraphPad Prism 9.0.0 was used for the statistical analysis. All experiments were performed in triplicate and the findings were presented as mean ± SD. In order to determine whether there was a significant difference, an unpaired two-tailed Student’s t-test was used, and statistical significance was defined as *p* < 0.05.

## 3. Results

### 3.1. Structure Identification of Isolated Compounds

As shown in Figure 1, twenty-one compounds, including three new compounds and eighteen known compounds, were acquired from the roots and rhizomes of *G. glabra*. The structures of the novel compounds were determined using NMR, high resolution mass spectrometry (HR-ESI-MS), CD and UV spectroscopy. Detailed information is available in the Appendix A and ^1^H-NMR and ^13^C-NMR data are described in Table 1.

Compound (**1**) had a molecular formula of C_25_H_22_O_6_, according to the [M-H]^−^ ion at *m*/*z* 417.1343 (calcd for C_25_H_21_O_6_, 417.1416) in the HR-ESI-MS and ^13^C-NMR data. The UV absorption maxima were at 247, 275 and 390 nm. The ^1^H and ^13^C-NMR signals at *δ*_H_ 8.23 (1H, s)/*δ*_C_ 155.7 (C-2), 120.7(C-3), 180.9 (C-4), and the HMBC correlations exhibited long-range cross-peaks from H-2 (*δ*_H_ 8.23) to *δ*_C_ 120.7 (C-3), which indicated an isoflavone skeleton (Figure 2). The appearance of two 2,2-dimethylpyran rings were established through ^1^H-NMR signals at *δ*_H_ 6.69 (1H, d, *J* = 10.0 Hz), 5.79 (1H, d, *J* = 10.0 Hz) and 1.45 (6H, s); *δ*_H_ 6.61 (1H, d, *J* = 10.0 Hz), 5.65 (1H, d, *J* = 10.0 Hz) and 1.31 (6H, s). One group was attached to C-7 and C-8 of ring A, according to the HMBC interactions of H-1″ (*δ*_H_ 6.69) with C-7 (*δ*_C_ 161.8), C-8 (*δ*_C_ 101.3), and C-9 (*δ*_C_ 159.2). Another was attached to C-3′ and C-4′ of ring B, according to the HMBC correlations of H-1‴ (*δ*_H_ 6.61) with C-2′ (*δ*_C_ 151.9), C-3′ (*δ*_C_ 109.5), and C-4′ (*δ*_C_ 152.1). Furthermore, two ortho-coupled protons signals at *δ*_H_ 6.44 (1H, d, *J* = 8.0 Hz), 6.93 (1H, d, *J* = 8.0 Hz), and two hydroxyl signals at *δ*_H_ 13.13 (s) and 9.88 (s) were detected in the ^1^H-NMR spectrum. The structure of compound **1** was similar to that of licoisoflavone D [17]. The isoprenyl group in licoisoflavone D was changed by a 2,2-dimethylpyran ring in compound **1**. Therefore, the structure of compound **1** was identified as 5,2′-dihydroxy-[2,2-dimethylpyrano(5,6:8,7)]-[2,2-dimethylpyrano(5,6:3′,4′)]isoflavone, and named licopyranol A.

Compound (**2**) had the molecular formula that was established to be C_25_H_28_O_6_ depending on its HR-ESI-MS data [M-H]^−^ ion at *m*/*z* 423.1814 (calcd for C_25_H_27_O_6_, 423.1886). The UV spectrum showed maximum absorption peaks at 290 and 369 nm. The ^1^H and ^13^C-NMR spectroscopical data showed that compound **2** had a isoflavanone skeleton, as proven by the typical signals corresponding to an oxygenated methylene and a hypomethylene at *δ*_H_ 4.50 (1H, dd, *J* = 10.8, 10.8 Hz), 4.45 (1H, dd, *J* = 5.6, 10.8 Hz) and 4.15 (1H, dd, *J* = 5.6, 10.8 Hz)/*δ*_C_ 70.6, an oxygenated carbon (*δ*_C_ 70.6), *δ*_C_ 47.2 and a carbonyl carbon (*δ*_C_ 198.5). In addition, two ortho-coupled protons signals at *δ*_H_ 6.31 (1H, d, *J* = 8.0 Hz) and 6.76 (1H, d, *J* = 8.0 Hz); two 2,2-dimethyl-dihydropyran rings at *δ*_H_ 2.55 (2H, t, *J* = 6.8Hz), 1.77 (2H, t, *J* = 6.8 Hz), 1.31(3H, s) and 1.28(3H, s); *δ*_H_ 2.48 (2H, t, *J* = 6.8 Hz), 1.68 (2H, t, *J* = 6.8 Hz), 1.16 (3H, s) and 1.13 (3H, s), and two hydroxyl signals at *δ*_H_ 12.11 (s) and 9.40 (s) were noticed in the ^1^H-NMR spectrum. One 2,2-dimethyl-dihydropyran ring was fused to C-7 and C-8, as evidenced by the HMBC cross-peaks (Figure 2) of H-1″ (*δ*_H_ 2.55)/C-7 (*δ*_H_ 162.1)/C-8 (*δ*_H_ 100.7)/C-9 (*δ*_H_ 161.4). Another was attached to C-3′ and C-4′, depending on the HMBC interactions of H-1‴ (*δ*_H_ 2.48) with C-2′ (*δ*_C_ 152.4), C-3′ (*δ*_C_ 108.9), and C-4′ (*δ*_C_ 155.7). To determine the conformation of C-3 in compound **2**, its CD spectrum was analyzed; however, there was no cotton effect, suggesting that compound **2** may be a racemic mixture. The two different configurations can be further purified by a chiral column [2]. Unfortunately, we failed to separate these due to the small amount of compound **2**. Therefore, the 2D structure of compound **2** was identified as 5,2′-dihydroxy-[2,2-dimethyl-3,4-dihydropyrano (5,6:8,7)]-[2,2-dimethyl-3,4-dihydropyrano(5,6:3′,4′)]isoflavanone, and named licopyranol B.

Compound (**3**) had the molecular formula of C_25_H_28_O_6_ that was determined by the HR-ESI-MS data ([M-H]^−^
*m*/*z* 423.1800, calcd for C_25_H_27_O_6_, 423.1886). The UV absorption maxima were at 240, 289 and 375 nm. Comparison of the NMR data of compound **3** with those of **2** (Table 1) showed their identical structural parts, except for the fact that the 2,2-dimethyl-dihydropyran in A-ring of compound **2** was replaced by an isoprenyl group (*δ*_H_ 3.10 (2H, d, *J* = 7.2 Hz), 5.10 (1H, t, *J* = 7.2 Hz), 1.67 (3H, s), and 1.62 (3H, s)) in compound **3**. The HMBC spectrum showed long-range cross-peaks (Figure 2) of H-1″ (*δ*_H_ 3.10) with C-7 (*δ*_C_ 164.9), C-8 (*δ*_C_ 107.3) and C-9 (*δ*_C_ 160.2); thus, the position of the isoprenyl was located at C-8. Similar to compound **2**, one pyran ring was bonded to C-3′ and C-4′ of ring B. Similarly, the CD spectrum of compound **3** indicated it was a racemic mixture and was not subjected to further purification due to the small amount. Hence, compound **3** was characterized as 5,7,2′-trihydroxy-8-(3-methyl-2-butenyl)-[2,2-dimethyl-3,4-dihydropyrano(5,6:3′,4′)] isoflavanone, and named licopyranol C.

In addition, based on the analysis of NMR spectra of compounds **4–21**, 18 known compounds were identified as follows: isoliquiritigenin (**4**) [18], glycyrol (**5**) [19], 6,8-diprenylgenistein (**6**) [20], lupiwighteone (**7**) [21], gancaonin L(**8**) [22], luteone (**9**) [23], 2,3-dehydrokievitone (**10**) [24], 7-O-methylluteone (**11**) [25], isochandalone (**12**) [26], isoderrone (**13**) [27], licoisoflavone B (**14**) [28], methyleriosemaone D (**15**) [29], euchestraflavanone A (**16**) [30], liquiritigenin (**17**) [21], glabrol (**18**) [31], dihydroformononetin (**19**) [32], licoisoflavanone C (**20**) [17], and 1-(4-ethenylphenyl)enthanone (**21**).

### 3.2. Inhibitory Effect of 21 Compounds on 6 Tumor Cell Lines

The antitumor activities of the extracts and isolated compounds were investigated. The extracts of petroleum ether, dichloromethane, and ethyl acetate layers showed better inhibitory effects on RKO and HT-29 cells, and their cell viability values were less than 40% at the concentration of 80 μg/mL for 48 h (Appendix A).

A total of 21 isolated compounds were tested for their inhibitory effect on human colorectal cancer RKO, HT-29 and HCT-116 cells, lung cancer A549 cells, liver cancer Huh7 cells and gastric cancer MGC-803 cells using MTT assay. As shown in Figure 3, when the tumor cells were processed with different quantities of compound-containing medium (10, 20, 40 μM) for 48 h, compounds **1**, **2**, **5**, **9**, and **12** showed better inhibitory effects in a dose-dependent manner. More importantly, some compounds inhibited tumor cells more strongly than 5-fluorouracil when administered at 40 μM, e.g., compounds **1**, **2**, **5**, and **12** better inhibited cells on HT-29, HCT-116 and Huh7, compound **1** better inhibited cells for A549 cells and compounds **1**, **5**, and **12** better inhibited cells for MGC-803. In addition, compound **9** had a better inhibitory effect on RKO, HT-29 cells and A549 cells in a concentration-dependent manner; compound **15** had a better inhibitory effect on lung cancer A549 cells in a dose-dependent manner, but had less or almost no inhibitory effect on the other five tumor cell lines. the significant inhibitory effect of compound **21** on lung cancer A549 cells in a concentration-dependent manner was also interesting, which was superior to 5-fluorouracil at a low concentration of 10 μM. The IC_50_ values of several compounds with better inhibitory effects are listed in Table 2.

### 3.3. Compounds ***1*** and ***5*** Inhibited the Proliferation of RKO Cells

Colorectal cancer (CRC) is a prevalent type of cancer and is the major cause of cancer deaths [33]. To further investigate the inhibitory effects of new compound **1** and the more effective compound **5** on the proliferation of human colorectal cancer RKO cells, we performed cell proliferation and cell cloning, in addition to cell cycle assays.

In cell proliferation experiments, we obtained growth curves for compounds **1** and **5** at different concentrations and for different times, as shown in Figure 4. The results showed that the obvious inhibitory effect of compound **1** on RKO cells occurred at a concentration of 20 μM at around 48 h. However, compound **5** showed a significant inhibitory effect at 10 μM, and the number of cells decreased continuously within 48 h. At around 72 h, the number of cells showed a weak increase, but much less than in the control group, probably because the pro-apoptotic effect of compound **5** on RKO cells changed to an inhibitory effect on their proliferation.

In cell cloning assays, as in Figure 5, both compounds **1** and **5** inhibited the formation of RKO cell clones and showed a dose-dependent effect. Compounds **1** and **5** inhibited the size and number of RKO cell clones when given at 5 μM, and the inhibition was particularly pronounced at 20 μM, with only some of the smaller clones or almost no clones present.

The cell cycle assays showed that both compounds **1** and **5** could produce a significant cycle blocking effect on RKO cells, as shown in Figure 6. Compound **1** was found to have a significant blocking effect on the G0-G1 stage of the cells when the RKO cells were processed with 10, 20, and 40 μM for 48 h, with a significantly raised percentage of cells in the G0-G1 phase, compared with control (from 53.49 ± 1.24% to 77.49 ± 2.98%); in addition, when the cells were treated with 10 μM, the percentage of cells in the G2-M increased significantly (from 14.36 ± 1.91% to 26.49 ± 1.35%). Compound **5** was found to have a significant blocking effect on the G2-M phase of cells (from 14.36 ± 1.91% to 25.69 ± 0.44%) after 48 h of treatment of RKO cells with 10 and 20 μM, while it blocked cells in the S phase when administered at concentrations up to 40 μM, with the percentage of cells in this period increasing from 32.14 ± 2.57% to 58.16 ± 2.78%. Thus, compound **1** blocked RKO cells in the G0-G1 and G2-M phases, while compound **5** blocked RKO cells in the S and G2-M phases.

### 3.4. Compounds ***1*** and ***5*** Promoted Apoptosis in RKO Cells

As shown in Figure 7, compounds **1** and **5** significantly induced apoptosis in RKO cells, and were potential dose-dependent apoptosis promoters, significantly increasing the rate of apoptosis compared to the control group (4.70 ± 0.40%, early and late apoptosis). Compound **1** increased apoptosis from 8.23 ± 0.29% to 33.42 ± 1.79% after treatment of RKO cells with 10, 20, and 40 μM for 48 h. Under the same conditions, compound **5** had a more significant apoptosis-promoting effect on RKO cells, with the apoptosis rate increasing from 8.14 ± 1.37% to 54.50 ± 0.67%. These results demonstrated that compounds **1** and **5** could promote apoptosis of RKO cells and inhibit RKO cells.

### 3.5. Compound ***5*** Inhibited RKO Cell Growth via the Wnt/β-Catenin Signaling Pathway

The Cancer Genome Atlas (TCGA) shows that the Wnt signaling pathway is activated in 93% of non-hypermutated CRC and 97% of hypermutated CRC [34]. Therefore, Wnt/β-catenin signaling is an appropriate drug goal with sufficient potency for the treatment of CRC. As shown in Figure 8, compounds **1** and **5** showed inhibitory effects on the Wnt/β-catenin signaling pathway. There was strong evidence that calmodulin plays a role in tumor development, invasion and metastasis. Furthermore, E-cadherin is a tumor main hallmark molecule that is significant in EMT progression [35]. Compounds **1** and **5** reduced E-cadherin expression after treatment of RKO cells with 10, 20 and 40 μM for 48 h. The effect of compound **5** was particularly pronounced and dose dependent, probably as a result of the synergistic effect of multiple signaling pathways. Over expression of β-catenin is a hallmark driver of the conventional Wnt pathway in colorectal cancer [36]. Compounds **1** and **5** both reduced the expression of β-catenin in RKO cells, and compound **5** was particularly effective in a dose-dependent manner. The expression of cancer-related c-Myc protein was slightly increased after treatment of RKO cells with compound **1**, while it was significantly reduced after treatment with compound **5** in a dose-dependent effect. The GSK-3β protein is a regulatory protein and its aggregation with Axin and other proteins will lead to the phosphorylation of β-catenin to p-β-catenin, which will lead to its degradation, and thus avoid nucleation and activation of downstream oncogenes. Both compounds reduced the expression of GSK-3β protein in RKO cells, and compound **5** had a more significant, dose-dependent effect.

## 4. Discussion

Licorice contains flavonoids, triterpenoid saponins, coumarins and stilbenoids [36], which are considered to have good biological activity [37]. Licorice flavonoids have been found to have anticancer activities with various underlying molecular pathways [7]. Licoricidin suppresses the progress of SW480 human colorectal carcinoma cells via stimulation of cycle arrest, apoptosis and the autophagic pathway [38]. Yan Lin et al. isolated 67 free phenolic compounds from licorice, and 11 of these compounds showed strong cytotoxic activity on 3 human cancer cell types (HepG2, SW480 and MCF7), whereas they demonstrated slight toxicity against the human normal cells of LO2 and HEK293T [39]. Isoprenyl-substituted chalcones had cytotoxic effects on MCF-7, HT-29 and A-2780 cancer cells, while the transformation of chalcones to flavonoids led to reduced anti-proliferative activity [40]. In addition, the isoprenyl flavonoid artonol A displayed cytotoxic activity against human lung cancer cells [41]. Moreover, natural products are considered as main sources of novel structures and discovery of new drugs, with about 48.6% of drugs actually being either natural products or directly derived therefrom [42]. Due to the widespread applications of licorice in traditional medicines, along with their recently reported bioactivities [43], we aimed to isolate and purify the compounds from the ethyl acetate extract of *G. glabra*.

In our study, the isolated compounds showed good antitumor activities, of which 10 compounds with isoprenyl or dimethylpyran rings showed better inhibition of tumor cells. In terms of the conformation–activity relationship, compounds 1 and 2 with two dimethylpyran ring substitutions were more cytotoxic to tumor cells than compound 3, with one isoprenyl and one dimethylpyran ring substitution, suggesting that the conversion of the isoprenyl group to the dimethylpyran ring in this class of compounds may be more favorable to its enhanced antitumor activity. In addition, compound 15, after deisoprenylation and methoxy substitution by compound 12, showed a weak inhibitory effect on tumor cells.

Colorectal cancer (CRC) is a serious threat to human health. More than 94% of colorectal cancer situations have mutations in one or more Wnt/β-catenin signaling pathway compositions [44]. Compounds **1** and **5**, which were significantly cytotoxic to colorectal cancer RKO cells, were selected for further studies of the anti-cancer mechanism. Our results showed that both compounds **1** and **5** could inhibit the growth of RKO cells by inhibiting cell proliferation, reducing clone formation and promoting their apoptosis, and compound **5** showed a superior effect. It is well known that the Wnt/β-catenin signaling pathway is closely associated with the progression of CRC [45]. Based on Western blotting analysis, compound **5** significantly reduced the expression of E-cadherin, β-catenin, c-Myc and GSK-3β proteins in RKO cells in a dose-dependent manner. It has been demonstrated that glycyrol stimulated cell death related with apoptosis and autophagy in AGS and HCT 116 cells, and inhibited tumor growth in a nude mouse tumor xenograft model bearing HCT 116 cells [46]. The benzofuranyl, isopentenyl and methoxy groups present in glycyrol played an important role in its anti-cancer activity, whereas the furan group led to more improvements [47]. Many natural products play a role in the Wnt/β-catenin signaling pathway [48], such as green tea polyphenols, epigallocatechin-3-gallate and resveratrol [49]. Lonchocarpin flavonoids acted as a Wnt/β-catenin pathway inhibitor, which has both in vitro and in vivo inhibitory effects on cell migration and cell proliferation on HCT116, SW480, and DLD-1 colorectal cancer cell lines [50]. Anticancer bioactive peptide (ACBP) also showed inhibition of proliferation, migration, and cell invasion in three CRC lines (HCT116, RKO, HT29) by suppressing the canonical Wnt signaling pathway [51].

Compound **1** decreased the expression of E-cadherin, β-catenin and GSK-3β proteins, but caused a weak increase in the expression of c-Myc protein. Therefore, the main signaling pathway through which compound **1** exerts its inhibitory effect on RKO cells needs to be further investigated.

## 5. Conclusions

In conclusion, we performed phytochemical studies on the roots of *G. glabra* and isolated 21 compounds, including 3 new compounds, further enriched their chemical composition and screened several compounds with better antitumor activities, which contained isoprenyl and dimethylpyran ring substitutions. Licopyranol A (**1**) and glycyrol (**5**) suppressed the growth of RKO cells and inhibited proliferation of RKO cells, reduced clone formation and promoted their apoptosis. The Western blotting results showed that glycyrol could inhibit the growth of RKO cells via the Wnt/β-catenin signaling pathway, suggesting that it had greater potential for antitumor activity, which is expected to become an anticancer seed compound through further studies.

## Figures and Tables

**Figure 1 metabolites-12-00896-f001:**
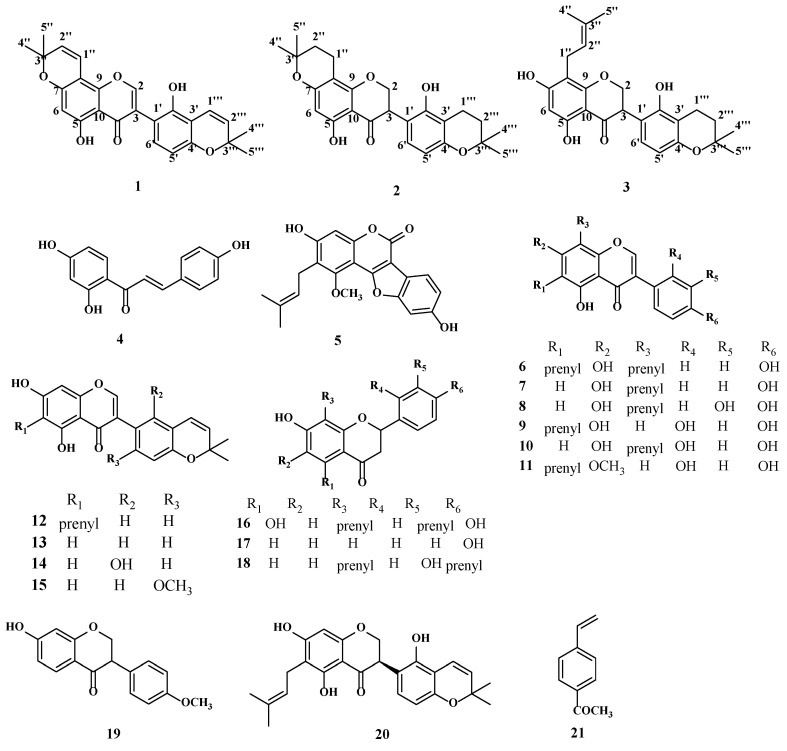
Structures of compounds **1**–**21**.

**Figure 2 metabolites-12-00896-f002:**
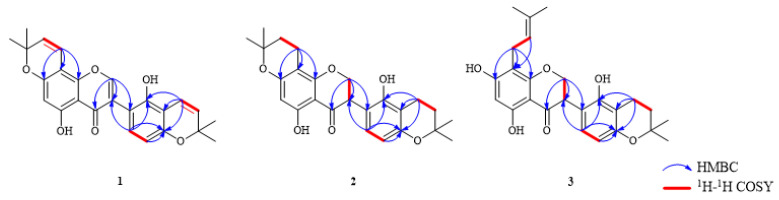
Key HMBC and ^1^H-^1^H COSY correlations for compounds **1**–**3**.

**Figure 3 metabolites-12-00896-f003:**
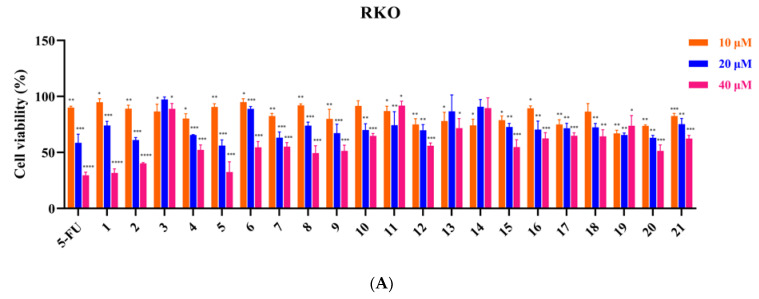
The effect of 21 compounds on 6 tumor cell lines (**A**–**F**) treated with 10, 20 and 40 μM for 48 h. Percent cell viability was expressed as mean ± SD. * *p* < 0.05, ** *p* < 0.01, *** *p* < 0.001, **** *p* < 0.0001, in comparison to control groups.

**Figure 4 metabolites-12-00896-f004:**
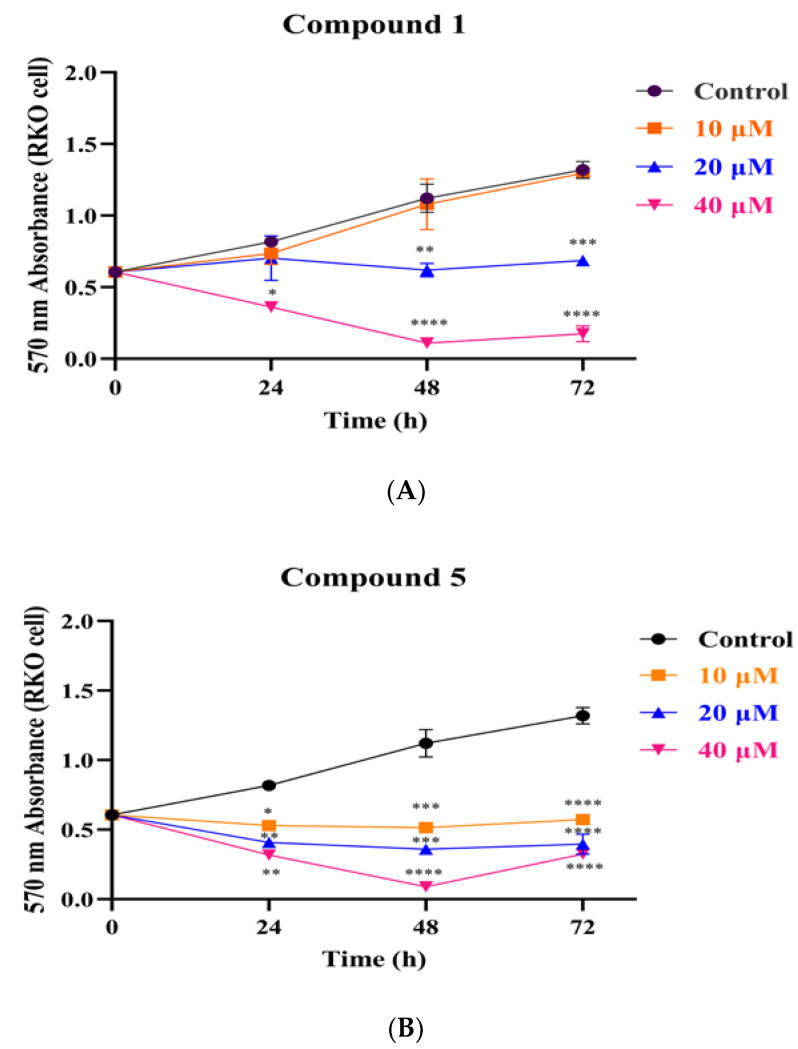
The growth curves of compounds **1** (**A**) and **5** (**B**) after treatment of RKO cells with 10, 20 and 40 μM for 0, 24, 48, and 72 h. * *p* < 0.05, ** *p* < 0.01, *** *p* < 0.001, **** *p* < 0.0001, in comparison with the corresponding standard groups.

**Figure 5 metabolites-12-00896-f005:**
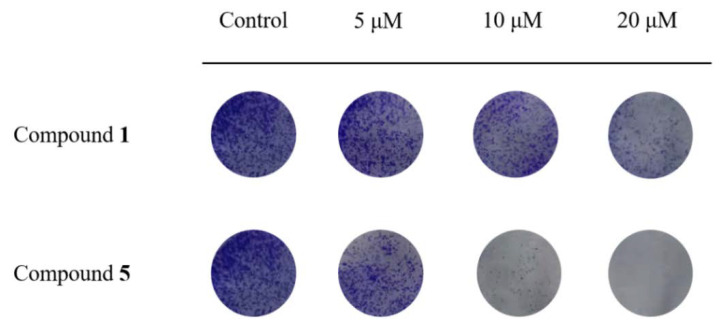
The effects of compounds **1** and **5** on RKO cell clones after 7 days of treatment with 5, 10 and 20 μM.

**Figure 6 metabolites-12-00896-f006:**
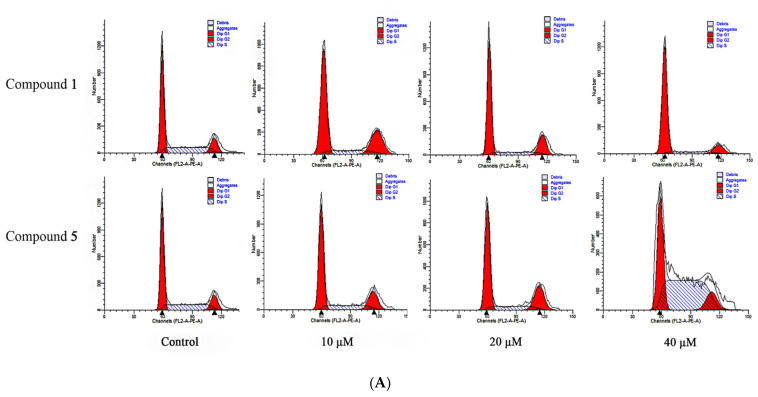
The effects of compounds **1** and **5** on the RKO cell cycle. (**A**) Cells were processed with 10, 20 and 40 μM for 48 h, respectively, collected, then stained with PI and analyzed by flow cytometry and the results were processed with Modfit LT 5.0; (**B**) statistics of RKO cell cycle results.

**Figure 7 metabolites-12-00896-f007:**
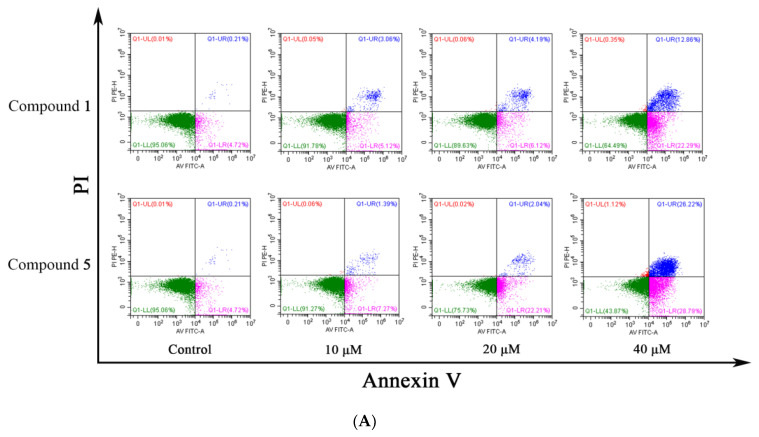
Apoptosis of RKO cells were induced by compounds **1** and **5**. (**A**) Cells were treated with 10, 20 and 40 μM for 48 h, respectively, then collected and dyed with Annexin V/PI and analyzed by flow cytometry. (**B**) Statistics of RKO cell apoptosis results. * *p* < 0.05, ** *p* < 0.01, *** *p* < 0.001, **** *p* < 0.0001, comparing with standard groups.

**Figure 8 metabolites-12-00896-f008:**
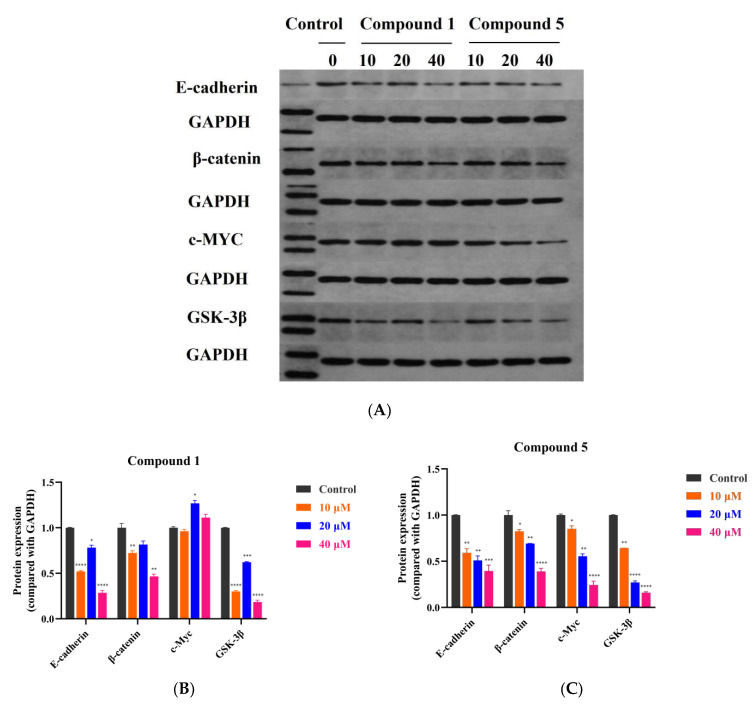
The effects of compounds **1** and **5** on the Wnt/β-catenin signaling pathway in RKO cells. (**A**) Expression of E-cadherin, β-catenin, c-Myc and GSK-3β proteins were determined by Western blotting after cells were treated with 10, 20 and 40 μM for 48 h, respectively. (**B**,**C**) Statistics of Western blotting results. * *p* < 0.05, ** *p* < 0.01, *** *p* < 0.001, **** *p* < 0.0001, compared with control groups.

**Table 1 metabolites-12-00896-t001:** NMR spectroscopic data for compounds **1**–**3** in DMSO-*d*_6_ (*δ* values in ppm, *J* values in Hz).

Position	1		2		3	
	*δ* _H_	*δ* _C_	*δ* _H_	*δ* _C_	*δ* _H_	*δ* _C_
2α	8.23, s	155.7	4.50, dd (10.8,10.8)	70.6	4.41, dd (10.4, 10.4)	70.5
2β			4.45, dd (10.8, 5.6)		4.38, dd (10.4, 6.4)	
3		120.7	4.15, dd (10.8, 5.6)	47.2	4.09, dd (10.4, 6.4)	47.2
4		180.9		198.5		198.3
5		154.0		160.4		162.0
6	6.27, s	99.9	5.85, s	96.8	5.98, s	95.9
7		161.8		162.1		164.9
8		101.3		100.7		107.3
9		159.2		161.4		160.2
10		105.7		103.0		102.6
1′		109.7		113.6		114.0
2′		151.9		152.4		152.3
3′		109.5		108.9		108.9
4′		152.1		155.7		155.6
5′	6.44, d (8.0)	107.8	6.31, d (8.0)	106.1	6.29, d (8.0)	106.0
6′	6.93, d (8.0)	131.7	6.76, d (8.0)	128.7	6.75, d (8.0)	128.6
1″	6.69, d (10.0)	114.4	2.55, t (6.8)	16.2	3.10, d (7.2)	21.6
2″	5.79, d (10.0)	128.7	1.77, t (6.8)	31.6	5.10, t (7.2)	123.5
3″		78.7.		76.5		130.6
4″	1.45, s	28.2	1.31, s	27.4	1.62, s	25.9
5″	1.45, s	28.2	1.28, s	26.5	1.67, s	18.1
1‴	6.61, d (10.0)	117.2	2.48, t (6.8)	17.4	2.51, t (6.8)	17.4
2‴	5.65, d (10.0)	129.1	1.68, t (6.8)	31.9	1.70, t (6.8)	31.9
3‴		76.3		74.3		74.2
4‴	1.31, s	27.9	1.16, s	27.3	1.15, s	27.4
5‴	1.31, s	27.9	1.13, s	25.9	1.12, s	25.9

**Table 2 metabolites-12-00896-t002:** IC_50_ values (μM) of several compounds with better inhibitory effects.

Compound	RKO	HT-29	HCT-116	A549	Huh7	MGC-803
**5-FU**	28.2 ± 0.7	25.7 ± 2.9	29.8 ± 5.8	46.4 ± 10.5	33.6 ± 5.2	3.3 ± 0.3
**1**	31.4 ± 0.5	28.0 ± 1.5	28.3 ± 0.9	25.3 ± 0.4	26.5 ± 1.5	24.0 ± 0.2
**2**	32.1 ± 0.2	28.0 ± 1.4	31.2 ± 1.5	>40	29.4 ± 0.4	26.1 ± 1.6
**5**	29.1 ± 3.1	11.9 ± 1.8	20.6 ± 7.3	>40	24.3 ± 1.5	13.5 ± 4.3
**6**	>40	>40	25.8 ± 1.7	32.3 ± 3.8	>40	25.4 ± 1.3
**12**	>40	27.5 ± 0.2	31.1 ± 6.6	26.5 ± 3.1	34.4 ± 5.2	23.5 ± 0.6
**15**	>40	>40	>40	31.13 ± 5.2	>40	>40
**21**	>40	>40	>40	11.5 ± 6.9	>40	>40

## Data Availability

The data used to support the findings of this study are available from the corresponding author upon reasonable request.

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
