# Peer review of "Three New Compounds, Licopyranol A–C, Together with Eighteen Known Compounds Isolated from Glycyrrhiza glabra L. and Their Antitumor Activities"

_metabolites, 2022, doi:10.3390/metabo12100896_

Round 1

Reviewer 1 Report

The manuscript entitled "Three New Compounds, Licopyranol A-C, Together with Eight-Een Known Compounds Isolated from Glycyrrhiza glabra L. and Their Antitumor Activities" by Wang et al. aimed to isolate three novel compounds with 18 previously known compounds were extracted from G. glabra, and the inhibitory activities of these compounds were examined using six different types of tumor cells.

A research must give a critical analysis of the data. In other words, you must determine if published results withstand scientific examination. You must clearly describe your precise goals and objectives to do the above. As a result, you must acquire a critical evaluation of state of the art in your research. This is a necessary component of every article.

In spite of the widespread enthusiasm for investigating unexplored natural resources in the search for new drugs, there are several issues that call into question the impact and soundness of the current investigation.

Firstly, it's not clear why it makes sense to separate some compounds from only the methanolic and ethyl acetate extract of roots and rhizomes of G. glabra for their Antitumor activities. 

The title can be modified to "Antitumor activities of three new compounds Licopyranol A-C and Eighteen known compounds Isolated from Glycyrrhiza glabra L against different tumor cells".

The study's rationale should be justified what has already been reported on Glycyrrhiza glabra and what further prompted the authors to do the current study design.

Why the ultrasonic extraction was chosen, and what were the extraction conditions?

Can the yield of the tested compounds support possible future pharmacological applications?

The purity of the isolated compounds was not determined, but the authors considered the tested samples as pure compounds.

I would recommend revising the discussion section with a clear interpretation of the results with the help of recent references to justify the findings.

Typos and inaccuracies should be checked and corrected.

Author Response

Responses to reviewers

Dear Editors and Reviewers:

Thank you very much for giving us an opportunity to revise our paper entitled "Three New Compounds, Licopyranol A-C, Together with Eighteen Known Compounds Isolated from Glycyrrhiza glabra L. and Their Antitumor Activities" (metabolites-1880436). Your comments were very helpful and have improved the overall quality of the paper.

The manuscript has been carefully revised, and the revisions were highlighted in red in the manuscript. The point-by-point answers to the comments and suggestions were listed as follows.

Response to Reviewer 1:

Q1: A research must give a critical analysis of the data. In other words, you must determine if published results withstand scientific examination. You must clearly describe your precise goals and objectives to do the above. As a result, you must acquire a critical evaluation of the state of the art in your research. This is a necessary component of every article.

Response: Thank you very much for your comments. We have carefully revised the manuscript. In 4. Discussion part, we added some sentences, such as “Licorice flavonoids have been found to have anticancer activities with various underlying molecular pathways”. “Due to the widespread applications of licorice in traditional medicines along with the recently known bioactivities, we aimed to isolate and purify the compounds from ethyl acetate extract of G. glabra.”.

Q2: In spite of the widespread enthusiasm for investigating unexplored natural resources in the search for new drugs, there are several issues that call into question the impact and soundness of the current investigation.

Response: Thank you very much. In lines 647-650, we have added “Moreover, the natural products consider as main sources of novel structures and discovery of new drugs, about 48.6% of drugs actually being either natural products or directly derived therefrom.”.

Q3: Firstly, it's not clear why it makes sense to separate some compounds from only the methanolic and ethyl acetate extract of roots and rhizomes of G. glabra for their Antitumor activities.

Response: We really appreciate your question. According to previous studies, acetate extract contain more flavonoids than other parts. In the furture, we will isolate the other metabolites from the other extracts.

Q4: The title can be modified to "Antitumor activities of three new compounds Licopyranol A-C and Eighteen known compounds Isolated from Glycyrrhiza glabra L against different tumor cells".

Response: Thank you very much for your suggestion. In the first half of the manuscript, we mainly focused on the isolation and identification of compounds isolated from licorice, then we evaluated their antitumor activities. Thus, we want to remain the title of “Three New Compounds, Licopyranol A-C, Together with Eighteen Known Compounds Isolated from Glycyrrhiza glabra L. and Their Antitumor Activities”.

Q5: The study's rationale should be justified what has already been reported on Glycyrrhiza glabra and what further prompted the authors to do the current study design.

Responses: Sincerest thanks for your valuable comment. In the introduction part, line 34-36, we have mentioned that “G. glabra is also one of the most commonly used herbs in the world, which possesses hepatoprotective, anti-inflammatory, neuroprotective, antioxidant, and antiviral activities.”. We also added some sentences of “Therefore,Glycyrrhiza glabra, and other natural anticancer agents have been popular studied due to that these compounds are considered with better bioactivities and may improve the side effect of cancer therapy.” in Line 47-50.

Q6: Why the ultrasonic extraction was chosen, and what were the extraction conditions?

Response: Thank you very much. Ultrasonic extraction is a simple and fast extraction technique for crud drugs. In this technique, a smashed sample is mixed with the suitable solvent and placed into the ultrasonic bath, while temperature and extraction time are controlled and enhances the mass transport by disrupting membranes, plant cells walls and other structures. We have added the extraction conditions of “under 45 kHz at 45°C” in 2.3 Extraction and isolation part.

Q7: Can the yield of the tested compounds support possible future pharmacological applications?

Response: Thanks for your valuable comments. We think these compounds have other biological activities. We hope to conduct meticulous research on these compounds in the future. We also have demonstrated that “which is expected to become an anticancer seed compound through further studies.” in the conclusions.

Q8: The purity of the isolated compounds was not determined, but the authors considered the tested samples as pure compounds.

Response: Thank you very much. We used TLC under three different eluents to identify the purity of the compounds. Then we also confirmed the purity of the isolated compounds by NMR.

Q9: I would recommend revising the discussion section with a clear interpretation of the results with the help of recent references to justify the findings.

Response: Thank you very much for your comments. We have added a recent references to justify our findings in the revised manuscript. In discussion part we have added “Lonchocarpin flavonoid acted as a Wnt/β-catenin pathway inhibitor, which has both in vitro and in vivo inhibitory effects on cell migration and cell proliferation on HCT116, SW480, and DLD-1 colorectal cancer cell lines. Anticancer bioactive peptide (ACBP) also showed inhibitions of proliferation, migration, and cell invasion in three CRC lines (HCT116, RKO, HT29) by suppressing the canonical Wnt signaling pathway.

Q10: Typos and inaccuracies should be checked and corrected.

Response: Thank we have checked and corrected the typos and inaccuracies in the manuscript highlighting in red.

Finally, we appreciate your time in reviewing our manuscript and the reviewers for their valuable suggestions and comments. We hope our responses satisfy you and the reviewers.

Sincerely yours,

Zhigang Yang

Reviewer 2 Report

Please see the pdf attachment.

Author Response

Responses to reviewers

Dear Editors and Reviewers:

Thank you very much for giving us an opportunity to revise our paper entitled "Three New Compounds, Licopyranol A-C, Together with Eighteen Known Compounds Isolated from Glycyrrhiza glabra L. and Their Antitumor Activities" (metabolites-1880436). Your comments were very helpful and have improved the overall quality of the paper.

The manuscript has been carefully revised, and the revisions were highlighted in red in the manuscript. The point-by-point answers to the comments and suggestions were listed as follows.

Response to Reviewer 2:

Q1: In the manuscript:

Compound 1[M-H]- ion at m/z 417.1343

Compound 2[M-H]- ion at m/z 423.1813

Compound 3[M-H]- m/z 423.1813

Therefore ,according to HR-ESI-MS extracted ion chromatograms in supplementary Figure ,compound 1 to 3 have m/z 417.1343 (RT 0.315), m/z 423.1814 (RT 0.318) , m/z 423.1814 (RT 0.318) respectively .

Why do compounds 2 and 3 have the same HR-ESI-MS data figure? Could you please provide a new chromatogram with a clear image?

Responses: Sincerest thanks for your valuable comment. We have checked the MS data and revised these data in the manuscript and supplementary material. Compound 1 to 3 showed m/z 417.1343 (RT 0.315), m/z 423.1814 (RT 0.318) , m/z 423.1800 (RT 0.329), respectively.

 Q2:  Please make a new Figure in supplementary material to zoom in on some of the near carbon peaks in Figures S5, S13, and S21.

Responses: we really appreciate your observation. We have modified these Figures in the supplementary material.

Q3:  All cosy words in supplementary material should be capitalized .

Responses: Thank you, we have corrected it in the supplementary material.

Q4:  Please spell in correctly of HR-ESI-MS in the supporting materials.

Responses: We appreciate your observation. HRESIMS was corrected to HR-ESI-MS in the supporting materials.

Q5:  Please increase the resolution of Figure 1.

Responses: Thank you very much. Figure 1 has been modified.

Q6:   Please correct the space in 5,2′-dihydroxy-[2,2-dimethylpyrano (5,6:8,7)]-[2,2-dimethylpyrano (5,6:3′,4′)]isoflavone

Responses: Thank you. It has been corrected in the revised manuscript

Q7: Please correct ‘Eight-Een’ in the tittle.

Responses: We really appreciate your observation. But we did not find the error in the title.

Q8: L221 , correct the space in (calcd for C25H27O6 , 423.1886) .

Responses: Thank you. It has been corrected in the revised manuscript.

Q9:  Figure 4, it should be ‘control’ instead of ‘contro’ .

Responses: Thank you very much. “Contro” was corrected to “control” in the revised manuscript.

Q10:  The supporting materials should be made available online.

Responses: Thank you. We have submitted the supporting materials online.

Finally, we appreciate your time in reviewing our manuscript and the reviewers for their valuable suggestions and comments. We hope our responses satisfy you and the reviewers.

Sincerely yours,

Zhigang Yang

Round 2

Reviewer 1 Report

I appreciate the efforts of the authors in the revision; however, I do not agree with the answers of the questions raised in the first review report

1) Firstly, it's not clear why it makes sense to separate some compounds from only the methanolic and ethyl acetate extract of roots and rhizomes of G. glabra for their Antitumor activities.

Authors have written in their response, "According to previous studies, acetate extract contain more flavonoids than other parts. In the furture, we will isolate the other metabolites from the other extracts."

2) It is still unclear to separate some compounds from only the methanolic and ethyl acetate extract of roots and rhizomes of G. glabra for their Antitumor activities.

If the ethyl acetate extract contains more Flavanoids than the other extract, it should be mentioned in the manuscript text with the proper references to make the rationale clear for the readers. So what about the selection of methanolic extract for Isolation? 

I agree that G. glabra is one of the most commonly used herbs in the world and is well studied for many pharmacological activities along with anticancer effects. So what was already reported on ethyl acetate and methanolic, root and rhizome extract in the aspect of isolation and anti-cancer effects? All these points should be discussed in the introduction before the current study design aims to provide a clear research gap.

3) It is unclear why ultrasonic extraction was chosen compared to other reported methods in G. glabra isolation studies. Have you done this extraction the first time and then isolated the compounds? 

4) Along with a clear Rf value, melting point, UV lambda Max and hydrolysis of compounds showed the purity of an isolated compound obtained from the column.

Author Response

Dear Editors and Reviewers:

Thank you very much for giving us an opportunity again to revise our paper entitled "Three New Compounds, Licopyranol A-C, Together with Eighteen Known Compounds Isolated from Glycyrrhiza glabra L. and Their Antitumor Activities" (metabolites-1880436). Your comments were very helpful and have improved the overall quality of the paper.

The manuscript has been carefully revised, and the revisions were highlighted in red in the manuscript. The point-by-point answers to the comments and suggestions were listed as follows.

Response to Reviewer 1:

Q1: Firstly, it's not clear why it makes sense to separate some compounds from only the methanolic and ethyl acetate extract of roots and rhizomes of G. glabra for their Antitumor activities.

Authors have written in their response, "According to previous studies, acetate extract contain more flavonoids than other parts. In the furture, we will isolate the other metabolites from the other extracts."

Response: We really appreciate your question. In lines 50-62, we added some sentences, such as “It has been reported that licorice extracts exhibited cytotoxic effects on various cancer cells, 70%methanol licorice extract inhibited the proliferation of human breast cancer cell MCF7 and hepatocellular carcinoma cell HepG2. Flavonoids of licorice showed important inhibitory effects on colorectal, breast, prostate, liver, stomach, bladder, and lung cancers. Furthermore, the isolated compounds from methanolic and ethyl acetate extract of G. glabra were found to show significant cytotoxic and anticancer properties.

In this study, the extracts of G. glabra showed inhibitory activities on RKO and HT-29 cells, the ethyl acetate extract had better effects than other layers at 20 μg/ml (Figure S1).”.

Q2: It is still unclear to separate some compounds from only the methanolic and ethyl acetate extract of roots and rhizomes of G. glabra for their Antitumor activities.

If the ethyl acetate extract contains more Flavanoids than the other extract, it should be mentioned in the manuscript text with the proper references to make the rationale clear for the readers. So what about the selection of methanolic extract for Isolation?

I agree that G. glabra is one of the most commonly used herbs in the world and is well studied for many pharmacological activities along with anticancer effects. So what was already reported on ethyl acetate and methanolic, root and rhizome extract in the aspect of isolation and anti-cancer effects? All these points should be discussed in the introduction before the current study design aims to provide a clear research gap.

Response: Thank you very much. In lines 50-62, we added some sentences, such as “In recent years, biological activities especially antitumor and cytotoxic properties of licorice extracts and their isolated compounds have received much attention. The bioactive components of licorice have shown antitumor properties in both in vivo and in vitro studies. It has been reported that licorice extracts exhibited cytotoxic effects on various cancer cells, 70%methanol licorice extract inhibited the proliferation of human breast cancer cell MCF7 and hepatocellular carcinoma cell HepG2. Flavonoids of licorice showed important inhibitory effects on colorectal, breast, prostate, liver, stomach, bladder, and lung cancers. Furthermore, the isolated compounds from methanolic and ethyl acetate extract of G. glabra were found to show significant cytotoxic and anticancer properties.

In this study, the extracts of G. glabra showed inhibitory activities on RKO and HT-29 cells, the ethyl acetate extract had better effects than other layers at 20 μg/ml (Figure S1).”.

Q3:  It is unclear why ultrasonic extraction was chosen compared to other reported methods in G. glabra isolation studies. Have you done this extraction the first time and then isolated the compounds?

Response: Thank you very much. In lines 87-90, we have added “Ultrasound extraction method has become more popular due to its various features such as low energy consumption, less extraction time, less active compound degradation, suitable for thermo-sensitive compounds and high extraction yield. This method was applied before for the isolation of secondary metabolites from licorice.”

Q4:  Along with a clear Rf value, melting point, UV lambda Max and hydrolysis of compounds showed the purity of an isolated compound obtained from the column.

Response: Thank you very much for your questions. We have added the Rf value, and UV lambda Max data in Supplementary material (Table S1 Rf value and UV lambda Max of Compounds 4-21). Moreover, the structure and purity have been conformed by NMR methods, if we measure the melting point, we will lose the weight of isolated compounds. We need more weight of isolated compounds for the bioactive assays. It is also not necessary to hydrolyze the isolated compounds, as we did not isolate glycosides.

Finally, we appreciate your time in reviewing our manuscript and the reviewers for their valuable suggestions and comments. We hope our responses satisfy you and the reviewers.

Sincerely yours,

Zhigang Yang

Reviewer 2 Report

The authors have revised the manuscript adequately and thoroughly in response to my comments. Therefore, I can recommend publication in its current form.

Author Response

Dear Editors and Reviewers,

We would like to thank you very much for your final acception decision of our manuscript entitled "Three New Compounds, Licopyranol A-C, Together with Eighteen Known Compounds Isolated from Glycyrrhiza glabra L. and Their Antitumor Activities" (metabolites-1880436).

Thank you again for the kinds. 

Best wishes, 

Zhigang Yang